# Discontinuation of Glycopeptides in Patients with Culture Negative Severe Sepsis or Septic Shock: A Propensity-Matched Retrospective Cohort Study

**DOI:** 10.3390/antibiotics9050250

**Published:** 2020-05-13

**Authors:** Yong Chan Kim, Jung Ho Kim, Jin Young Ahn, Su Jin Jeong, Nam Su Ku, Jun Yong Choi, Joon-Sup Yeom, Yoon Soo Park, Young Goo Song, Ha Yan Kim

**Affiliations:** 1Department of Internal Medicine, Yonsei University College of Medicine, Seoul 03722, Korea; amomj@yuhs.ac (Y.C.K.); qetu1111@yuhs.ac (J.H.K.); jsj@yuhs.ac (S.J.J.); smileboy9@yuhs.ac (N.S.K.); seran@yuhs.ac (J.Y.C.); JOONSUP.YEOM@yuhs.ac (J.-S.Y.); YSPARKOK2@yuhs.ac (Y.S.P.); IMFELL@yuhs.ac (Y.G.S.); 2Department of Medicine, Yonsei University College of Medicine, Seoul 03722, Korea; 3Biostatistics Collaboration Unit, Yonsei University College of Medicine, Seoul 03722, Korea; hykim1213@yuhs.ac

**Keywords:** glycopeptides, antimicrobial stewardship, sepsis, septic shock, propensity score

## Abstract

Implementation of antibiotic stewardship is difficult in patients with sepsis because of severity of disease. We evaluated the impact of glycopeptide discontinuation (GD) in patients with culture negative severe sepsis or septic shock who received glycopeptides as initial empiric antibiotic therapy at admission. We conducted a single center retrospective cohort study between January 2010 and March 2018. GD was defined as discontinuation of initial empiric glycopeptides on availability of culture results, revealing the absence of identified pathogens. In 92 included patients, the leading causes of sepsis were pneumonia (34.8%) and intra-abdominal infection (23.9%); 28-day mortality and overall mortality were 14% and 21%, respectively. Glycopeptides were discontinued in 42/92 patients. After propensity score matching, baseline characteristics were not significantly different between the GD and non-GD (GND) groups. GND was associated with development of acute kidney injury (OR 5.54, 95% CI 1.49–20.6, *P* = 0.011). GD did not increase the 7-day, 14-day, and 28-day mortality compared with GND. The length of hospital stay was shorter in the GD group than in GND group (16.33 ± 17.11 vs. 25.05 ± 14.37, *P* = 0.082), though not statistically significant. GD may be safe and reduce adverse events of prolonged antibiotic use in patients with culture negative severe sepsis or septic shock receiving glycopeptides as initial empiric antibiotic therapy.

## 1. Introduction

International guidelines recommend the prompt use of empirical broad-spectrum antibiotics in patients with severe sepsis or septic shock [1]. Once the pathogen is identified and its susceptibility profile is determined, the use of empiric antibiotics should be adjusted to minimize inappropriate exposure to broad-spectrum antimicrobial agents [2]. This strategic de-escalation (DE) reduces the spectrum of antimicrobials by using narrow spectrum antibiotics, decreasing the number of antimicrobials in combination therapy, and discontinuing the use of unnecessary agents [3]. A study reported that DE therapy did not increase mortality risk in patients with severe sepsis or septic shock [4]. Another study even demonstrated improved survival in patients who received antimicrobial DE therapy [5]. However, only a few studies deal with the impact of DE therapy in relation to specific pathogens or specific antibiotics [6].

Glycopeptides are effective against infections caused by Gram-positive pathogens, such as enterococci or methicillin-resistant *Staphylococcus aureus* (MRSA). They are also used for empirical antimicrobial therapy in patients with severe sepsis and septic shock when MRSA infection is suspected [1]. Since the methicillin resistance rate of *Staphylococcus aureus* reached approximately 66% in South Korea [7], most clinicians select glycopeptides as initial empiric antibiotics for patients with severe sepsis or septic shock. However, many of them do not implement DE even when the initial microbiologic results do not suggest MRSA infection. Such excessive and inappropriate antimicrobial use has influenced the emergence of antimicrobial resistant organisms and increased adverse drug events [8,9]. In clinical practice, physicians’ decisions to discontinue the use of glycopeptides depend on the severity of a patient’s condition rather than the microbiologic results.

The absence of an identified pathogen was shown to be a risk factor for not performing antimicrobial DE, and many studies on antimicrobial DE excluded patients with negative culture results [3]. Therefore, it is necessary to study the impact and safety of glycopeptide discontinuation (GD) in critically ill patients with infection when there are no identified pathogens in the initial culture data. Therefore, in this study, we evaluated the impact of GD on the clinical outcomes in patients with severe sepsis or septic shock who received glycopeptides as initial empiric antibiotics and had negative culture results at admission in the emergency department.

## 2. Results

### 2.1. Study Population

Of the 1514 patients enrolled in the sepsis critical pathway program, only 111 patients with culture negative severe sepsis or septic shock, who received glycopeptides as the initial empirical antibiotics therapy, were eligible. Based on the exclusion criteria, an additional 19 patients were excluded. Therefore, 92 patients were finally included in the final analysis. GD was performed in 42 patients, and GD and non-GD (GND) in 50 patients (Figure 1).

Table 1 shows the clinical characteristics and outcomes of patients included in this study. The mean age of patients was 61.07 ± 14.26; 39.13% were women and 56.52% had an MRSA risk factor at admission. The most common underlying comorbidity was hypertension (54.34%), followed by cancer (41.3%) and diabetes mellitus (28.26%). The initial sequential organ failure assessment (SOFA) score (8.68 ± 2.86) was higher than the SOFA score on day 5 (4.1 ± 4.25). Approximately two-thirds of the patients (63.04%) were admitted to the intensive care unit (ICU) from the emergency department. Pneumonia accounted for the largest proportion of primary focus of sepsis (34.78%). During hospitalization, acute kidney injury (AKI) was developed in 17.39% of patients; 28-day mortality occurred in 15.22% and the overall mortality was 22.83%.

### 2.2. Characteristics of Patients Before and After Propensity Score Matching

Before propensity score matching (PSM), GND patients had higher lactate levels (4.9 ± 4.18 mmol/L vs. 3.3 ± 2.41 mmol/L, *P* = 0.029) and initial SOFA scores (9.36 ± 3.17 vs. 7.95 ± 2.37, *P* = 0.02) than GD patients (Table 2). In addition, the percentage of patients with congestive heart failure was significantly higher in the GND group than in the GD group (12% vs. 0%, *P* = 0.03). Although not statistically significant, the white blood cell counts (11,727.14 ± 10,485.66/mm^3^ vs. 16,539.4 ± 21,144.89/mm^3^), total bilirubin (0.91 ± 0.76 mg/dL vs. 1.22 ± 1.33 mg/dL), C-reactive protein (118.5 ± 87.18 mg/L vs. 152.78 ± 127.66 mg/L), SOFA score on day 5 (3.5 ± 3.62 vs. 4.7 ± 4.67), and ICU admission rate (57.1% vs. 68%) were higher in the GND group than in the GD group. Risk factors for MRSA were more frequently observed in the GD group than in the GND group (64.3% vs. 50%).

After PSM, 21 patients remained in each group. Significant differences in baseline characteristics disappeared between both propensity score matched groups. Comparing with patients in the GND group, those in the GD group had numerically lower C-reactive protein levels (127.84 ± 93.94 mg/L vs. 167.4 ± 152.37 mg/L, *p* = 0.339) and initial SOFA scores (8.05 ± 2.20 vs. 9.14 ± 3.73, *p* = 0.169). The percentage of patients with lung disease was numerically higher in the GD group than in the GND group. In addition, the overall distribution of the primary focus of sepsis was not significantly different between the two groups, but the proportion of septic focus that may require empirical glycopeptides, such as pneumonia, intra-abdominal infection, and surgical site infection was numerically higher in the GD group (76.2% vs. 57.1%, *P* = 0.102, data not shown).

### 2.3. Outcomes

The GD group had a shorter duration of glycopeptide use than GND group (4.39 ± 1.76 days vs. 14.86 ± 6.58 days, *P* = < 0.001). GND was significantly associated with a likelihood of AKI development both before matching (odds ratio (OR) 4.16, 95% confidence interval (CI) 1.49–11.6, *P* = 0.007) and after matching (OR 5.54, 95% CI 1.49–20.6, *P* = 0.011) (Table 3). In the GD group, none of the patients underwent hemodialysis after the use of glycopeptides, whereas three patients in the GND group underwent hemodialysis. Although the 7-day, 14-day, and 28-day mortalities were not significantly associated with glycopeptides DE before and after PSM, the 28-day mortality was numerically lower in the GD group than in the GND group before matching (9.5% vs. 20%, *p* = 0.172) and after matching (9.5% vs. 19.1%, *p* = 0.427). The length of hospital stay in the GD group was significantly shorter than that in the GND group before PSM (17.79 ± 18.65 days vs. 27.62 ± 21.71 days, *P* = 0.023). After PSM, although there was no statistical significance, the GD group showed a shorter length of hospital stay compared to that in the GND group. (16.33 ± 17.11 days vs. 25.05 ± 14.37 days, *P* = 0.082) (Table 3).

## 3. Discussion

In this propensity-matched retrospective cohort study, we demonstrated the outcomes of GD in patients with culture negative severe sepsis or septic shock Although PSM was performed with a small number of variables, the differences in baseline characteristics were not observed between the GD and GND groups after PSM. Mortality at 7, 14, and 28-days did not increase in patients in the GD group compared with those in the GND group. GND was associated with development of AKI, particularly in patients who underwent hemodialysis.

Proper management of patients with sepsis requires not only rapid recognition of the bacterial infection and administration of empirical broad-spectrum antibiotics, but also identification of the suspected causative bacteria. Furthermore, there is need to switch to narrow spectrum antibiotics based on the susceptibility results, and discontinuation of antibiotic therapy is important after use for an appropriate duration [1]. As rapid recognition and prompt antibiotic use should be guaranteed in patients with sepsis, many of them receive broad-spectrum antibiotics as the initial therapy more than necessary. When the clinical specimen culture results become available in patients with sepsis, empirical glycopeptides can be de-escalated or discontinued if MRSA is not identified [4]. Some physicians continue glycopeptides in patients when signs of infection persist even after negative culture results are identified. Our study showed that this practice did not improve clinical outcomes in patients; rather, it increased the development of AKI. The proportion of vancomycin used in this study was 40% and 31.5% in the GD group and the GND group, respectively (data not shown). The duration of vancomycin use is well known as one of the important determinants of vancomycin-induced renal toxicity [10]. The longer duration of glycopeptide use, including vancomycin, in the GND group may have contributed to the development of AKI. In addition, the higher incidence of AKI seems to serve as one of the reasons for the longer hospital stay in the GND group, although not statistically significant. Similar results were observed in another study, in which GD did not increase the mortality but reduced the incidence of AKI and total hospital length of stay in patients with culture negative nosocomial pneumonia [9].

This study did not include six cases of urinary tract infection in culture negative patients who received glycopeptides as initial antibiotics. There are limited data to recommend the empirical use of glycopeptides for treatment of urinary tract infection since MRSA is uncommon pathogen in urinary tract infection [11]. Looney and colleagues reported that a very low proportion of all urine samples tested were methicillin-resistant [12]. Our institution also does not recommend the use of glycopeptides for treatment of urinary tract infection, even in severe sepsis or septic shock, unless the causative strain is identified as MRSA. Therefore, we excluded cases of urinary tract infection that received initial empiric glycopeptides.

Variable selection is important for the study using the PSM. However, there is lack of consensus to determine which variables should be included in PSM. It is recommended to include as many variables as possible. However, since a small number of patients were included in this study, we performed the PSM using three variables (lactate, SOFA score on day 5, and congestive heart failure) that seemed to have the greatest influence on the decision of GD and mortality. Lactate is a well-known biomarker that can discriminate patients who are likely to die from those who are likely to live [1,13], and we previously showed that the lactate level was an independent prognostic factor in our cohort patients with severe sepsis and septic shock [14]. SOFA score at the time of the GD, day 5, in this study, would be more important than SOFA score at admission, and initial SOFA score was not likely to affect the physician’s decision for GD. Physicians tend to maintain the use of initial empirical antibiotics if the severity of the patients’ condition does not improve, regardless of the culture results. In fact, the mortality rate was significantly associated with the severity score on the day of available culture results for severe sepsis or septic shock in studies of antibiotics DE [5,15]. We were able to clearly demonstrate the impact of GD by adjusting the SOFA score at the time of GD. Although there was a difference in the initial SOFA score at admission between the GD and GND groups, the difference disappeared after PSM. Congestive heart failure was the only underlying comorbidity, in which a significantly different proportion was observed between the two groups. Furthermore, sepsis accounts for a large proportion, almost a quarter, of death in patients with congestive heart failure [16]. Therefore, we selected the lactate level, SOFA score on day 5, and congestive heart failure as the three variables for propensity matching. Our previous study showed that pneumonia and intra-abdominal infection had the highest mortality rates [14]. Although we did not include these variables in the PSM, there were no differences between the GD and GND groups before and after the PSM.

Few studies evaluated the impact of antibiotic DE in only culture negative patients with severe infections [9,17]. Antibiotic DE according to the identified pathogens and susceptibilities can improve the outcome of patients [3,18]. Therefore, antibiotic DE is easily performed if pathogens are identified in culture results [19]. We demonstrated that GD was safe and effective even if causative pathogens were not identified. To the best of our knowledge, our study included the largest number of culture negative severe sepsis or septic shock patients to evaluate the impact of GD. These findings can help clinicians implement antibiotic DE in patients with culture negative severe sepsis or septic shock.

Our study has several limitations. First, since a small number of patients were included in this study, the results should be cautiously interpreted because there were remaining potential bias and confounding factors that could affect the outcomes. In addition, although significant differences were not noted between the two groups after the PSM, we did perform PSM with a few variables due to the limited patient number. Further studies are warranted with a larger patient size. Second, this study was performed in a single center. Our institution’s large cohort consisted of adequate patients with severe sepsis or septic shock to represent this population in other hospitals in South Korea. However, the results observed in this study may not be applicable to other countries. Finally, there is a lack of data regarding the incidence of drug resistant organisms or *Clostridium difficile* infection after treatment. Excessive use of antibiotics causes a selection of those pathogens [20]. Because of the small sample size and short duration of observation, it was not suitable to confirm the occurrence of antibiotic resistant organisms.

## 4. Materials and Methods

### 4.1. Study Design and Population

A retrospective cohort study was performed at Severance Hospital (Seoul, South Korea), a 2400-bed tertiary teaching hospital at Yonsei University College of Medicine. A sepsis critical pathway program has been implemented in this institution to recognize patients with sepsis early and to begin prompt treatment. When adult patients (age ≥18 years) with suspected or proven infection visited the emergency department, those who had two or more systemic inflammatory response syndrome criteria and one of hypotension or high lactate level (≥4 mmol/L), previously defined as severe sepsis and/or septic shock, were included in the program [21,22]. The clinical management consisted of fluid resuscitation for tissue perfusion, empirical broad-spectrum antibiotic therapy, and vasopressor use in patients who remained hypotensive despite fluid resuscitation.

We analyzed the electronic medical records of patients registered in the sepsis critical pathway program from January 2010 to March 2018. Patients who received glycopeptides as initial empirical antibiotics were eligible for inclusion. Subsequently, we screened patients with no identified causative pathogen in the final culture results. In addition, we considered patients with nonpathogenic colonizers or normal flora in the culture results as having no identified causative pathogen. All patients were included in this study only for the first episode. The following cases were excluded in this study: (1) transfers to another hospital or hopeless discharge from hospital within 28 days after admission; (2) death within 7 days after initial culture; (3) identification of causative pathogen in the follow up culture; (4) glycopeptide use again within 7 days after discontinuation; (5) infections that may require continued anti-MRSA treatment, such as infective endocarditis, endovascular graft infection, mycotic aneurysm, device-related infection, and osteoarticular infection; (6) hospital acquired infection transferred from another hospital; and (7) urinary tract infection.

Patients were classified into two groups, the GD group and the glycopeptide non-discontinuation (GND) group. GD group was defined as the group in which initial empiric glycopeptides were discontinued when culture results became available and revealed the absence of identified pathogens. The opposite cases were defined as the GND group. Glycopeptides were designated antibiotics in the antimicrobial stewardship program at the hospital and were targets of preauthorization throughout the study period [23].

This study was approved by the institutional review board of the Yonsei University Health System Clinical Trials Center, and the protocol adhered to the tenets of the Declaration of Helsinki. Since the study was retrospective and the study participants were anonymized, the institutional review board waived the requirement for written consent from the patients.

### 4.2. Data Collection and Definition

Medical records of all patients were reviewed by one trained researcher and one infectious disease specialist. Data for baseline characteristics, laboratory results, radiologic findings, treatments, and outcomes were obtained from electronic medical records. Patients’ underlying comorbidities were defined using the International Classification of Diseases, 10th revision. Acute kidney injury (AKI) was identified according to the Kidney Disease: Improving Global Outcomes definition [24]. AKI was classified as stage 1, 2, and 3 as follows: (1) Stage 1, increase in serum creatinine ≥1.3 mg/dL or ≥ 1.5 times than baseline; (2) Stage 2, increase of ≥ 2 times than baseline; and (3) Stage 3, increase of ≥ 3 times than baseline. If the AKI progressed and hemodialysis was performed due to AKI, it was defined as “Hemodialysis needed”. Patients with underlying end stage renal disease were excluded from the evaluation of AKI development. The sequential organ failure assessment (SOFA) score was used to assess the severity in patients with severe sepsis or septic shock at admission. Since the severity of illness appeared to change during hospitalization, we re-evaluated the SOFA score on day 5, when the culture results were almost confirmed, to reassess the severity. The primary focus of sepsis was defined according to the Centers for Disease Control and Prevention/National Healthcare Safety Network surveillance criteria [25]. Patients who satisfied one or more of the following conditions were considered to have an MRSA risk factor: more than 2 days of hospitalization within the previous 90 days; living in a nursing facility; receipt of hemodialysis; infection with human immunodeficiency virus; receipt of antibiotic therapy for more than 3 days within 30 days; and intravenous drug use.

### 4.3. Statistical Analysis 

Categorical variables were expressed as numbers and percentages and were compared using Chi-square tests. Continuous variables were expressed as median and interquartile range (IQR) and were compared using the independent t-test (normal distribution) or Mann–Whitney test (non-normal distribution). To adjust the effects of potential confounding factors between the GD and GND subjects, a propensity score matching (PSM) analysis was used. Considering the total number of patients expected to be included in our study and to avoid overfitting in the matching, propensity scores were calculated through logistic regression model based on the following clinically selected factors: lactate level, SOFA score on day 5, and congestive heart failure. PSM was performed using the SAS macro. We used conditional logistic regression to compare the AKI and mortality between the GD and GND groups in the matched data. Comparison of length of hospital stay between groups, linear mixed model was performed in the matched data. Statistical analyses were performed using SAS version 9.4 (SAS Institute Inc., Cary, NC, USA).

## 5. Conclusions

Antibiotic DE can be one of the management strategies to avoid an increase in the mortality rate in patients with sepsis, while reducing adverse drug events and preventing antibiotic resistance. This study suggests that GD may be a reasonable option for management of patients with culture negative severe sepsis or septic shock. Further studies with large sample size are needed to support our results, and in terms of infection control, the effect of GD on preventing antibiotic resistance in such settings should be analyzed.

## Figures and Tables

**Figure 1 antibiotics-09-00250-f001:**
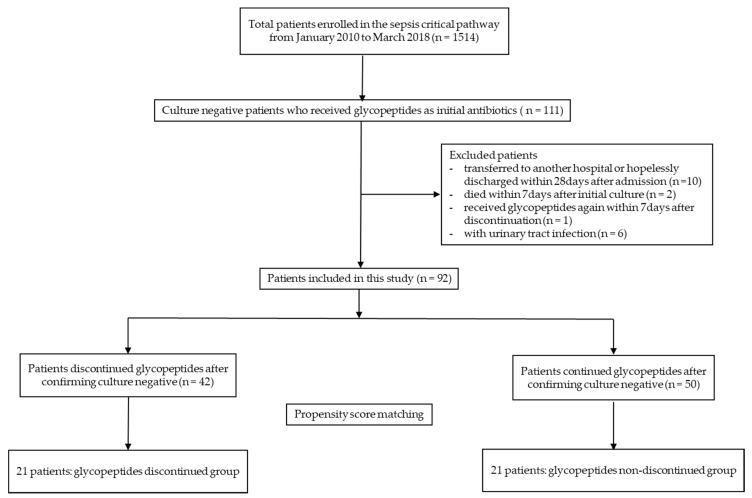
Flowchart of patients’ inclusion.

**Table 1 antibiotics-09-00250-t001:** Clinical characteristics and outcomes of patients in the study.

Variables	Total (*n* = 92)
Demographic	
Female, no. (%)	36(39.13)
Age, years	61.07(14.26)
Body mass index, kg/m^2^	23.96(10.98)
Laboratory	
White blood cell count per mm^3^	14,520(16,826)
Hematocrit, %	34.2(8.26)
Platelet per mm^3^	183,745(123,401.5)
Blood urea nitrogen, mg/dL	32.71(25.63)
Creatinine, mg/dL	2.31(2.38)
Total bilirubin, mg/dL	1.07(1.08)
Albumin, g/dL	3.18(0.74)
C-reactive protein, mg/L	137.58(112.75)
Lactate, mmol/L	4.12(3.18)
Comorbidities	
Congestive heart failure, no. (%)	6(6.52)
Hypertension, no. (%)	50(54.34)
Pulmonary disease, no. (%)	7(7.61)
Liver disease, no. (%)	6(6.52)
Diabetes mellitus, no. (%)	26(28.26)
Renal disease, no. (%)	13(14.13)
Cancer, no. (%)	38(41.3)
Risk factor for MRSA, no. (%)	52(56.52)
Initial SOFA score	8.68(2.86)
Admission to ICU from emergency department, no. (%)	58(63.04)
SOFA score at day 5	4.1(4.25)
Primary focus of sepsis	
Primary sepsis, no. (%)	19(20.65)
Pneumonia, no. (%)	32(34.78)
Intra-abdominal, no. (%)	22(23.91)
Skin and soft tissue, no. (%)	8(8.7)
Others ^a^, no. (%)	11(11.96)
Outcomes	
Acute kidney injury, no. (%)	16(17.39)
28-day mortality, no. (%)	14(15.22)
Overall mortality, no. (%)	21(22.83)

MRSA, methicillin-resistant *Staphylococcus aureus*; SOFA, sequential organ failure assessment; ICU, intensive care unit. ^a^ Others include central nervous system infections, gastroenteritis, deep neck infection, and others.

**Table 2 antibiotics-09-00250-t002:** Clinical characteristics of patients in both groups before and after propensity score matching.

Variables	Before Propensity Score Matching			After Propensity Score Matching		
	Glycopeptides Discontinued Group (*n* = 42)	Glycopeptides Non-Discontinued Group (*n* = 50)	*p*-Value	Glycopeptides Discontinued Group (*n* = 21)	Glycopeptides Non-Discontinued Group (*n* = 21)	*p*-Value
Demographic						
Female, no. (%)	17(40.48)	19(38)	0.809	9(42.86)	8(38.10)	0.706
Age, years	61.02 ± 12.9	61.12 ± 14.62	0.974	61.38 ± 13.75	63.14 ± 14.35	0.714
Body mass index, kg/m^2^	24.59 ± 16.22	23.39 ± 3.39	0.656	26.24 ± 23.1	23.21 ± 2.84	0.557
Laboratory						
White blood cell count per mm^3^	11,727.14 ± 10,485.66	16,539.4 ± 21,144.89	0.161	13,250.48 ± 10,956.11	13,380 ± 12,844.07	0.968
Hematocrit, %	34.08 ± 7.47	35.26 ± 8.62	0.489	33.95 ± 8.05	32.98 ± 7.55	0.696
Platelet per mm^3^	179,238.1 ± 131,796.19	185,660 ± 122,677.05	0.81	151,428.57 ± 87,436.59	189,380.95 ± 123,681.64	0.239
Blood urea nitrogen, mg/dL	32.35 ± 33.84	32.97 ± 18.04	0.915	34.18 ± 39.22	30.75 ± 17.1	0.725
Creatinine, mg/dL	2.05 ± 2.43	2.46 ± 2.44	0.421	2.4 ± 3.19	2.38 ± 2.89	0.981
Total bilirubin, mg/dL	0.91 ± 0.76	1.22 ± 1.33	0.168	0.97 ± 0.91	0.86 ± 0.53	0.642
Albumin, g/dL	3.19 ± 0.72	3.17 ± 0.78	0.93	3.11 ± 0.72	3.27 ± 0.65	0.419
C-reactive protein, mg/L	118.5 ± 87.18	152.78 ± 127.66	0.138	127.84 ± 93.94	167.4 ± 152.37	0.339
Lactate, mmol/L	3.3 ± 2.41	4.9 ± 4.18	0.029	3.23 ± 2.08	3.04 ± 1.96	0.576
Comorbidity						
Congestive heart failure, no. (%)	0(0)	6(12)	0.03	0(0)	0(0)	NA
Hypertension, no. (%)	23(54.76)	27(54)	0.942	12(57.14)	13(61.90)	0.739
Pulmonary disease, no. (%)	4(9.52)	3(6)	0.698	2(9.52)	0(0)	NA
Liver disease, no. (%)	3(7.14)	3(6)	>0.999	1(4.76)	1(4.76)	>0.999
Diabetes mellitus, no. (%)	11(26.19)	15(30)	0.686	5(23.81)	6(28.57)	0.706
Renal disease, no. (%)	5(11.9)	8(16)	0.574	3(14.29)	3(14.29)	>0.999
Cancer, no. (%)	18(42.86)	20(40)	0.782	9(42.86)	9(42.86)	>0.999
Risk factor for MRSA, no. (%)	27(64.29)	25(50)	0.169	11(52.38)	11(52.38)	>0.999
Initial SOFA score	7.95 ± 2.37	9.36 ± 3.17	0.02	8.05 ± 2.20	9.14 ± 3.73	0.169
Admission to ICU from ED, no. (%)	24(57.14)	34(68)	0.283	13(61.90)	13(61.90)	>0.999
SOFA score at day 5	3.5 ± 3.62	4.7 ± 4.67	0.178	3.33 ± 3.5	3.71 ± 4.6	0.527
Primary focus of sepsis			0.801			0.609
Primary sepsis, no. (%)	9(21.43)	10(20)		3(14.29)	6(28.57)	
Pneumonia, no. (%)	16(38.10)	16(32)		6(28.57)	4(19.05)	
Intra-abdominal, no. (%)	10(23.81)	12(24)		9(42.86)	4(19.05)	
Skin and soft tissue, no. (%)	2(4.76)	6(12)		1(4.76)	4(19.05)	
Others ^a^, no. (%)	5(11.9)	6(12)		2(9.52)	3(14.29)	

MRSA, methicillin-resistant *Staphylococcus aureus*; SOFA, sequential organ failure assessment; ICU, intensive care unit; ED, emergency department. ^a^ Others include central nervous system infections, gastroenteritis, deep neck infection, and others.

**Table 3 antibiotics-09-00250-t003:** Comparison of clinical outcomes between groups before and after propensity score matching.

Outcomes	Before Propensity Score Matching						After Propensity Score Matching				
	Glycopeptides Discontinuation Group (*n* = 42)	Glycopeptides Non-Discontinuation Group (*n* = 50)	OR	95% CI	*p*-Value		Glycopeptides Discontinuation Group (*n* = 21)	Glycopeptides Non-Discontinuation Group (*n* = 21)	OR	95% CI	*p*-Value
New AKI, no. (%)			4.16	1.49–11.6	0.007				5.54	1.49–20.6	0.011
None	38(90.1)	37(74)					20(95.2)	16(76.2)			
1-Creatinine ≥1.3 mg/dL or ≥ 1.5 times than baseline	0(0)	4(8.16)					0(0)	1(5)			
2-≥ 2 times than baseline	2(4.76)	1(2.04)					1(4.76)	0(0)			
3-≥ 3 times than baseline	1(2.38)	1(2.04)					0(0)	0(0)			
Hemodialysis needed	1(2.38)	6(12.24)					0(0)	3(15)			
7-day mortality, no. (%)	1(2.38)	1(2)	0.84	0.05–13.8	0.901		1(4.76)	1(4.76)	1	0.06–15.99	>0.999
14-day mortality, no. (%)	2(4.76)	6(12)	2.73	0.52–14.29	0.235		1(4.76)	1(4.76)	1	0.06–15.99	>0.999
28-day mortality, no. (%)	4(9.52)	10(20)	2.38	0.69–8.22	0.172		2(9.52)	4(19.05)	2	0.37–10.92	0.427
**Outcomes**	**Before propensity score matching**						**After propensity score matching**				
	**Glycopeptides discontinuation group** **(n = 42)**	**Glycopeptides** **Non-discontinuation group(n = 50)**	**β**	**Standard error**	***p*-value**		**Glycopeptides discontinuation group** **(n = 21)**	**Glycopeptides** **Non-discontinuation group** **(n = 21)**	**β**	**Standard error**	***p*-value**
Hospital day	17.79 ± 18.65	27.62 ± 21.71	9.834	4.264	0.023		16.33 ± 17.11	25.05 ± 14.37	8.714	4.876	0.082

AKI, acute kidney injury; OR, odds ratio; CI, confidence interval.

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
