# Peer review of "Discontinuation of Glycopeptides in Patients with Culture Negative Severe Sepsis or Septic Shock: A Propensity-Matched Retrospective Cohort Study"

_antibiotics, 2020, doi:10.3390/antibiotics9050250_

Round 1
Reviewer 1 Report
Thank you for giving the opportunity to review the paper. The paper is generally well written, but there is a concern the conclusion. From this study analysis, the result just present the probability of GD in culture negative population in severe sepsis or septic shock patient. This study is just retrospective observational study from less than 100 study population, not a clinical trial. So describing “antibiotic DE is a proper management strategy to avoid an increase in mortality...” is illogical. "No statisctical differences in two group” does not support GD in clinical practice. I think it is much better suggesting mildly.
Author Response
Thank you for giving the opportunity to review the paper. The paper is generally well written, but there is a concern the conclusion. From this study analysis, the result just presents the probability of GD in culture negative population in severe sepsis or septic shock patient. This study is just retrospective observational study from less than 100 study population, not a clinical trial. So describing “antibiotic DE is a proper management strategy to avoid an increase in mortality...” is illogical. "No statistical differences in two group” does not support GD in clinical practice. I think it is much better suggesting mildly.
Response: Thank you for your thoughtful comments. We have changed the sentences more logically and mildly according to your comments in conclusions, line 270-273, as below.
“Antibiotic DE can be one of the management strategies to avoid an increase in the mortality rate in patients with sepsis, while reducing adverse drug events and preventing antibiotic resistance. This study suggests that GD may be a reasonable option for management of patients with culture negative severe sepsis or septic shock.”

Reviewer 2 Report
I read the paper with interest and i think it is a well designed research discussing a clinal relevant topic.
1) I have some doubt about propensity score matching: have you treid tho other way, that is to say to retain all the possible explanatory variables in the model? Baseline clinical information could be loose by using the propensity score
2) can you prduce survival curves? Thanks
Author Response
Reviewer 2:
I read the paper with interest and I think it is a well-designed research discussing a clinical relevant topic.
1) I have some doubt about propensity score matching: have you tried to other way, that is to say to retain all the possible explanatory variables in the model? Baseline clinical information could be loose by using the propensity score
Response: Thank you for your thoughtful comments. We agree with you. Unfortunately, we had no choice but to perform propensity score matching with a few variables due to the limited patient number. A statistical expert, H.Y.K., advised to select three variables which were considered to be clinically most relevant. The candidates for the propensity score matching were congestive heart failure, lactate, initial SOFA score, and SOFA score at day 5, which seemed to have the greatest influence on the decision of GD or showed significant difference between GD and GND groups. SOFA score at the time of the GD, day 5, would be more important than initial SOFA score. In addition, to avoid overfitting in the matching, we finally excluded initial SOFA score for propensity score matching. Although we selected only three variables for propensity score matching, there were no significant differences in variables not selected as well as those selected between both groups.
2) can you produce survival curves? Thanks
Response: Thank you for your detailed comments. We have made the survival curve with Kaplan-Meier analysis as below.
Figure1. Kaplan-Meier plots of the 28-day survival in the patients stratified according to the glycopeptides discontinuation.
Figure2. Kaplan-Meier plots of the 28-day survival in the patients stratified according to the glycopeptides discontinuation after propensity score matching.
As above, there were no statistically significant differences between the groups. However, the number of mortality cases was too small to show in the manuscript.
